

# A New Reduction Model for Enhancing the Interpolation Accuracy of VMF1/VMF3 Tropospheric Products in GNSS Applications

Peng Sun[1], Kefei Zhang[1,2], Dantong Zhu[3], Dongsheng Zhao[1], Shuangshuang Shi[1,4], Xuexi Liu[1], Minghao Zhang[1], Suqin Wu[1]

[1] School of Environment Science and Spatial Informatics, China University of Mining and Technology, Xuzhou 221116, China

[2] Satellite Positioning for Atmosphere, Climate and Environment (SPACE) Research Centre, RMIT University, Melbourne VIC 3001, Australia

[3] College of Surveying and Geo-Informatics, North China University of Water Resources and Electric Power, Zhengzhou 450000, China

[4] Shandong Provincial Key Laboratory of Water and Soil Conservation and Environmental Protection, College of Resources and Environment, Linyi University, Linyi 276000, China

*Correspondence to*: Kefei Zhang (profkzhang@cumt.edu.cn); Peng Sun (peng_sun@cumt.edu.cn)

**Abstract.** Grid-wise Vienna Mapping Functions 1 (VMF1) and Vienna Mapping Functions 3 (VMF3) tropospheric products have been widely used to interpolate the a priori zenith hydrostatic delay (ZHD) and zenith wet delay (ZWD) over the GNSS (Global Navigation Satellite Systems) stations for the mitigation of tropospheric delays inherited in GNSS observations. Since the two products only provide ground surface ZHD and ZWD for global grid points, the ZHD and ZWD of the four grid points nearest to the GNSS site need to be reduced to the same height of the GNSS site before a horizontal interpolation (e.g., bilinear interpolation or inverse-distance weighted interpolation) is implemented. However, the accuracy of the officially recommended simple reduction model may not be as good as desired if the height of a GNSS site largely differs from that of the four ground surface grid points to be used in the interpolation. In this contribution, a new reduction model for each grid point is developed for reducing the grid-wise ZHD and ZWD to the target height to improve the interpolation performance. The sample data for the modelling were the 10-year (2010–2019) ZHD and ZWD profiles over the grid points obtained from ERA5 monthly-mean reanalysis data, while 3-year (2020–2022) ERA5 hourly reanalysis and IGS (International GNSS Service) site-wise ZTD products were used to evaluate the new model. Test results showed that the accuracy of the ZHD, ZWD (as well as the ZTD) interpolated from the VMF1/VMF3 products deduced by the new model was significantly better than the ones deduced by traditional methods, especially for sites with substantial height disparities from adjacent VMF grid points. It is expected that the new model adds good value to related fields such as GNSS positioning and GNSS meteorology for better performance.

## 1 Introduction

Tropospheric delay occurs when an electromagnetic wave is transmitted through the atmosphere due to the refraction of the neutral gas. The tropospheric delay (TD) is defined as:





$$TD = 10^{-6} \int N \, ds + \left( \int ds - L \right) \tag{1}$$

where the first and second term denote the delay resulting from the velocity change and geometric path change, respectively; $N$ is the refractivity index; $s$ is the transmission path, i.e., the bended path, of the signal; $L$ is the length of satellite-receiver straight line. The refractivity index $N$ is defined as:

$$
\begin{aligned}
N &= k_1 \cdot \frac{P_d}{T} + \left( k_2 \cdot \frac{e}{T} + k_3 \cdot \frac{e}{T^2} \right) \\
&= k_1 \cdot \frac{P}{T} + \left( k_2' \cdot \frac{e}{T} + k_3 \cdot \frac{e}{T^2} \right) \\
&= N_h + N_{nh}
\end{aligned}
\tag{2}
$$

where $k_1$, $k_2$, $k_3$ are three refractivity constants (Rüeger, 2002; Thayer, 1974); $P_d$ and $e$ are atmospheric pressures resulting from dry air and water vapor, respectively; $P$ is the total atmospheric pressure ( $P = P_d + e$ ); $T$ is the atmospheric temperature; $N_h$ and $N_{nh}$ are hydrostatic and non-hydrostatic part of the refractivity, respectively. $k_2'$ is a constant related to $k_1$ and $k_2$:

$$k_2' = k_2 - k_1 \frac{R_d}{R_v} \tag{3}$$

where $R_d$ and $R_v$ are the specific gas constants for dry air and water vapor, respectively. $R_d = R / M_d$, $R_v = R / M_v$, where $R$ is the universal gas constant (8.3143 J/K/mol$)$; $M_d$ (28.9644 g/mol) and $M_w$ (18.0152 g/mol) are the molar mass of dry air constant and water, respectively.

The tropospheric delay is one of the major error sources embedded in observations of space geodetic techniques, such as GNSS (Global Navigation Satellite Systems) and VLBI (Very Long Base-line Interferometry). Since the second term in (1) is quite smaller (usually less than 0.1 mm for elevation angles above 57°) than the first term, and the bending effect can be absorbed by advanced mapping functions (see (5)) (Möller and Landskron, 2019; Nafisi et al., 2012), researchers mainly focus on the modelling of the first term in (1). As a convention, the zenith tropospheric delay (ZTD) is widely used in GNSS and VLBI data processing:

$$ZTD = \int N_h \, dh + \int N_{nh} \, dh = ZHD + ZWD \tag{4}$$

where ZHD and ZWD are zenith hydrostatic delay and zenith non-hydrostatic delay (usually called zenith wet delay (ZWD)), respectively. Then a slant tropospheric delay can be modelled using ZHD, ZWD together with mapping functions (Chao, 1974; Chen and Herring, 1997; Niell, 1996):

$$TD = ZHD \cdot mf_h + ZWD \cdot mf_w + \Delta T_{grad} \tag{5}$$

where $mf_h$ and $mf_w$ are the mapping functions for the hydrostatic and non-hydrostatic part of the tropospheric delay, respectively; $\Delta T_{grad}$ is the tropospheric gradient, which is caused by the azimuthal asymmetry of the troposphere.





The modelling accuracy of ZHD affects the accuracy of GNSS-estimated station height and ZTD (Boehm et al., 2006; Tregoning and Herring, 2006; Kouba, 2009). Furthermore, an accurate ZHD is necessary for converting ZTD to precipitable water vapor (PWV) in GNSS meteorology (Bevis et al., 1992; Wang et al., 2017; Zhu et al., 2024). For ZWD, a pre-obtained ZWD can also be used for the direct correction of the wet delay or treated as a pseudo-observation for the constraining of the wet delay to accelerate the convergence of precise point positioning (PPP) (Sun et al., 2021a). Thus, it has been a continuous

global effort in improving the accuracy of ZTD for the data processing of space geodetic techniques.

The ZHD can be modeled with a desirable accuracy using Saastamoinen model with in-situ atmospheric pressure ($P$) measurements as input (Davis et al., 1985; Saastamoinen, 1972):

$$ZHD = \frac{0.0022768P}{f(\varphi, H)} = \frac{0.0022768P}{1 - 0.00266\cos(2\varphi) - 0.00028H} \tag{6}$$

where $\varphi$ and $H$ are the latitude (in radians) and height (in km) of the GNSS site, respectively. Although the ZWD can also be

calculated in a similar way, i.e., using an empirical model, e.g., the Askne-Nordius model, together with in-situ meteorological measurements, its accuracy is poor due to the dynamic nature of water vapor (Chen and Liu, 2016). The Askne-Nordius model is expressed as:

$$ZWD = 10^{-6}(k_2' + k_3 / T_m)\frac{R_d}{(\lambda + 1)g_m}e_s \tag{7}$$

where $T_m$ is the weighted mean temperature; $\lambda$ is the water vapor decay parameter; $g_m$ is the mean gravitational acceleration;

$e_s$ is the surface water vapor pressure.

Since most GNSS stations are not mounted with meteorological sensors and it is complex for real-time GNSS users to process forecasted NWM data to obtain the atmospheric parameters, empirical tropospheric delay models like UNB3m (Leandro et al., 2006) and GPT models are commonly used (Boehm et al., 2007; Böhm et al., 2015; Lagler et al., 2013; Landskron and Böhm, 2018). Considering the spatial-temporal variations of the meteorological variables, some advanced

models were developed to improve the modeling accuracy of the tropospheric delay (Huang et al., 2023; Jiang et al., 2024; Li et al., 2018; Sun et al., 2023; Yang et al., 2021b; Yao et al., 2015; Zhao et al., 2023). Such empirical models, while easy to use, have limited accuracy due to rapid variations of the troposphere (Wang et al., 2017; Xia et al., 2023).

Fortunately, the tropospheric delay can also be obtained from grid-wise Vienna Mapping Functions 1 (VMF1, with the resolution of 2.5°×2°) (Boehm et al., 2006, 2009) and Vienna Mapping Functions 3 (VMF3, with the resolution of 1°×1° and

5°×5°) (Landskron and Böhm, 2018) products provided by the Vienna University of Technology (TU Wien). The ZHD, ZWD, and the coefficients of VMF1 and VMF3 mapping functions for each point are determined at four epochs (0, 6, 12, 18 UTC) each day using NWM data from the European Centre for Medium-Range Weather Forecasts (ECMWF), and the tropospheric delay for the time and location of interest can be interpolated from its surrounding grid points. Two kinds of NWM data are used to generate the VMF1/VMF3 products, i.e., ECMWF OPERATIONAL NWM for VMF1_OP and

VMF3_OP, and ECMWF FORECAST NWM for VMF1_FC and VMF3_FC. In addition, some other VMF1-like products (GFZ-VMF1 and UNB-VMF1) are also publicly available for users (Santos, 2011; Zus et al., 2015).



Recent studies have shown that the accuracy of the tropospheric delay obtained from the grid-wise VMF1 and VMF3 products is significantly better than the ones predicted by empirical tropospheric delay models (Sun et al., 2021b; Yang et al., 2021a). Before the release of the VMF3 product, the VMF1 product was highly recommended for GNSS data processing (Kouba, 2008). Yao et al. (2018b) evaluated the ZTD predicted by VMF1_FC using references from IGS (International GNSS Service) final ZTD products and results demonstrated a 1.83 cm mean root-mean-square (RMS). Yuan et al. (2019) investigated the performance of VMF1_FC using real-time precise point positioning (PPP) and results showed that the accuracy of the PPP-estimated position and ZTD using VMF1_FC-based ZHD were better than those from empirical models. The VMF3 product, including two resolutions (1°×1° and 5°×5°), have become available since 2018, and the VMF3 mapping function outperforms VMF1's (Landskron and Böhm, 2018). Sun et al. (2021a) utilized the ZWD predicted by VMF3_FC (1°×1°) as a pseudo-observation to constrain the ZWD parameter in real-time single-frequency (SF) PPP and results indicated that the convergence time of SF-PPP was significantly shortened. Yang et al. (2021a) evaluated the accuracy of the ZHD and ZWD over China predicted by VMF3_OP using reference data from ERA5 reanalysis. Sun et al.(2021b) evaluated the accuracy of the ZHD predicted by VMF1_FC and VMF3_FC using 3-year surface atmospheric pressures measured at 443 globally distributed radiosonde stations and results showed that the mean RMSE of the ZHD values predicted by VMF1_FC, VMF3_FC (5°×5°) and VMF3_FC (1°×1°) at all the 443 stations were 5.9, 5.4, and 4.3 mm, respectively, and the accuracy of the ZHD predicted by VMF1_FC and VMF3_FC can meet the demands of real-time PWV retrieval.

To improve the accuracy of VMF1/VMF3-predicted ZTD, the officially recommended ZTD interpolation method was re-investigated, and then a new vertical reduction model was developed to reduce both ground surface ZHD and ZWD to the height of interest e.g., the height of a GNSS station, for an interpolated result from the four grid points surrounding the GNSS station. The methodology and data utilized in this research are presented firstly, followed by test results, their analyses, and discussion. Conclusions of this contribution are given in the final section.

## 2 Data Sources

### 2.1 ERA5 monthly mean reanalysis data

ERA5 reanalysis data are the state-of-the-art atmospheric reanalysis data provided by the ECMWF. In this contribution, the 10-year (2010–2019) ERA5 monthly mean geopotential, temperature and water vapor pressure reanalysis data at 37 pressure levels over the grid points of the VMF1 and VMF3 products were selected as the samples for the development of the new ZTD vertical reduction model. These geopotential heights were converted to ellipsoidal heights for adapting to geodetic applications following the equations given by Nafisi (2012).

### 2.2 ERA5 hourly reanalysis data

For the evaluation of the newly developed ZHD and ZWD vertical reduction models, ERA5 hourly reanalysis data over the grid points at 0, 6, 12, 18 UTC on 5th, 10th, 15th, 20th, 15th, and 30th day of each month in the 3-year period of 2020–2022 were selected for the calculation of the references. Since the horizontal resolution of the reference data coincides with the





VMF1 and VMF3 products, temporal interpolation and horizontal geospatial interpolation were not needed to carry out for the model evaluation.

**2.3 GNSS ZTD data**

ZTD products in the 3-year period of 2020–2022 at 394 IGS (International GNSS Service) stations were selected for the evaluation of the new ZHD and ZWD vertical reduction models. The geographical distribution of these stations is shown in

Fig. 1.

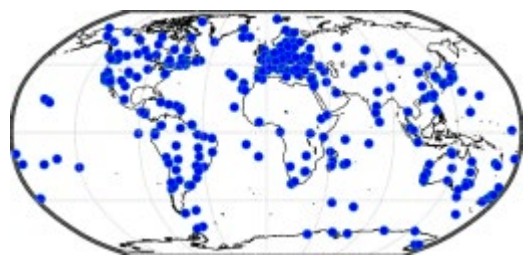

**Figure 1**. Geographical distribution of 394 GNSS stations selected for the evaluation of the new model

**3 Methodology**

**3.1 Re-investigation of officially recommended ZTD interpolation method**

To improve the accuracy of VMF1/VMF3-predicted ZTD, the officially recommended ZTD interpolation method, as follows, was re-investigated first:

[1]. Identifying the four ground grid points surrounding the target point S (see Fig. 2) provided in the VMF1 and VMF3 products, see points A~D. Then, for each of the four grid points, the following procedure is performed.

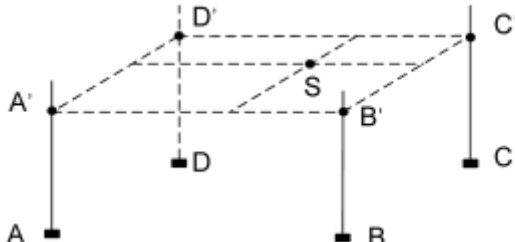

**Figure 2** Diagram for the interpolated ZHD and ZWD for the position of S based on the ZHD and ZWD values at the four ground grid points surrounding S (A to D) reduced to the height of S (*A'~D'*).

[2]. Performing a linear interpolation of the ZHD and ZWD values of the grid point in the temporal domain: data from the two neighbouring epochs (selected from 0, 6, 12, 18 UTC) that are most close to the GNSS observation time of S were used

in the interpolation, and the interpolated results were denoted by $ZHD_0$ in (8) and $ZWD_0$ in (10) since they are the ground surface values (i.e., A ~ D) of the grid point.





[3]. Using an inverse process of the Saastamoinen model to calculate the ground atmospheric pressure of the grid point:

$$P_0 = \frac{ZHD_0[1 - 0.00266\cos(2\varphi) - 0.00028h_0]}{0.0022768} \qquad (8)$$

where $P_0$ and $h_0$ are the ground atmospheric pressure and height of the grid point, respectively.

[4]. Reducing $P_0$ to the height of $S$ using the following vertical atmospheric pressure reduction model (Kouba, 2008):

$$P = P_0\left(1 - 0.0000266\Delta h\right)^{5.225} \qquad (9)$$

where $P$ is the reduced atmospheric pressure at the height of S (A'~D'); 0.0000266 is an empirical decay parameter for atmospheric pressure; $\Delta h$ is the difference between the target and reference heights.

[5]. Obtaining reduced ZHD (A'~D') using $P$ obtained in (9) and the Saastamoinen model.

[6]. Using product-provided ZWD, i.e., $ZWD_0$ and the following reduction model to obtain reduced ZWD (for A'~D'):

$$ZWD = ZWD_0 \bullet \exp\left(\frac{-\Delta h}{2000}\right) \qquad (10)$$

[7]. Repeating steps [2]-[6] for all the four points (A'~D'), then using the four reduced ZHD and ZWD values and the bilinear interpolation in the spatial domain to obtain the interpolated ZHD and ZWD for S.

### 3.2 GPT2w-based ZTD reduction method

It is noted that in (9) a fixed empirical atmospheric pressure decay parameter (0.0000266) was applied for the vertical reduction of atmospheric pressure on a global scale. However, the general state of atmospheric conditions varies with season in the temporal domain and also with region in the spatial domain (Wang et al., 2022; Zhang et al., 2021a). Thus, the decay parameter is better modelled dynamically in both spatial and temporal domain.

As is also shown in (9), since the vertical correction for atmospheric pressure is related to the difference between the reference and target heights, a low-accuracy decay parameter may significantly affect the accuracy of the reduced ZHD result. To address this issue, the atmospheric decay function utilized, i.e., (9), can be replaced with a better solution:

$$P = P_0\left(1 - \frac{\beta}{T_0}\Delta h\right)^{\frac{g_m \bullet M_d}{R \bullet \beta}} \qquad (11)$$

where $T_0$ is the temperature (in $K$) at the reference height; $\beta$ is the temperature lapse rate. The accuracy of these two variables affects the accuracy of the vertical correction of atmospheric pressure, thus these two variables or parameters need to be estimated properly. Fortunately, they can be predicted by empirical models like GPT2w, then (9) can be replaced with a temperature-dependent atmospheric pressure decay function, i.e., (11). Zhang et al. (2021b) utilized GPT2w-predicted atmospheric temperature and its lapse rate as the input of (11) for the vertical reduction to the ZHD provided by VMF1/VMF3-like products, and test results in the Tibetan Plateau region were significantly improved. In this research, the water vapor decrease factor ($\lambda$) predicted by GPT2w was adopted for the vertical reduction to the ZWD provided in grid-wise VMF1/VMF3 products using the ZWD decay function given by Dousa (2014):



$$ZWD = ZWD_0 \left(1 - \frac{\beta}{T_0}\Delta h\right)^{\frac{(\lambda+1)\cdot g_m}{R_d \cdot \beta}} \tag{12}$$

### 3.3 A new ZTD reduction method for VMF1/VMF3 products

If a constant temperature lapse rate ($\beta = 0.0065$ K/m) is utilized, the exponential term of (11) can be simplified as(Yao et al., 2018a):

$$P = P_0 \left[1 - \tau(h - h_0)\right]^{5.256} \tag{13}$$

where $\tau = \beta/T_0$. Since the denominator part of the Saastamoinen model, i.e., $f(\varphi, H)$ in (6), is approximately equal to 1, the specific value of the ZHD at height h above the reference height can be simplified as:

$$\frac{ZHD}{ZHD_0} = \frac{P_0 \left[1 - \tau(h - h_0)\right]^{5.256}}{P_0} \tag{14}$$

Then the ZHD at height h can be obtained by:

$$ZHD = ZHD_0 \left[1 - \tau(h - h_0)\right]^{5.256} \tag{15}$$

For ZWD, in this research, the ZWD decay function proposed by Dousa (2014) was modified to:

$$\frac{ZWD}{ZWD_0} = \left(\frac{P}{P_0}\right)^{\gamma+1} = \left[1 - \tau(h - h_0)\right]^{5.256(\gamma+1)} \tag{16}$$

Thus

$$ZWD = ZWD_0 \left[1 - \tau(h - h_0)\right]^{5.256(\gamma+1)} \tag{17}$$

In this contribution, $\tau$ and $\gamma$ for each of the VMF1/VMF3 grid points were modeled through the following steps:

[1]. For the spatial domain, $\tau$ and $\gamma$ at the ground surface of the grid point for the 120 months in the 10-year period of 2010–2019 were fitted using the ERA5 monthly-mean reanalysis data mentioned above.

[2]. For the temporal domain, the seasonal variations of $\tau$ and $\gamma$ at the grid point were modeled by fitting the above 120 monthly-mean $\tau$ and $\gamma$ values:

$$t = t_0 + A_1 \cos\left(\frac{DOY - d_1}{365.25}2\pi\right) + A_2 \cos\left(\frac{DOY - d_2}{365.25}4\pi\right) \tag{18}$$

where $t_0$ is the mean of the parameter (i.e., either $\tau$ or $\gamma$); $A_1$ and $A_2$ are the amplitudes of annual and semi-annual variation of the parameter, respectively; $d_1$ and $d_2$ are the day of year (DOY) corresponding to their initial phase. Then the ZHD and ZWD at height h can be obtained using (15) and (17), respectively.

### 4 Results and discussion

Three schemes for the reduction of the ground surface ZHD and ZWD values provided by grid-wise VMF1/VMF3 products were evaluated in this research, and they are:



1) Scheme 1 for the ZHD and ZWD resulting from officially recommended methods, i.e., (9) – (10);

2) Scheme 2 for the reduction functions expressed in (11) and (12) based on empirical temperature and its lapse rate, as

well as the water vapor decay parameter obtained from the GPT2w model

3) Scheme 3 for the new model based on (15) and (17).

The RMSE was used to measure the overall discrepancy and accuracy of the interpolated results obtained from the above three schemes:

$$RMSE = \sqrt{\frac{1}{n}\sum_{i=1}^{n}\left(ZTD_{VMF,i} - ZTD_{ref,i}\right)^2} \qquad (19)$$

where $ZTD_{VMF}$ and $ZTD_{ref}$ are the VMF-based ZTD (including ZHD and ZWD) and the reference ZTD, respectively.

It should be noted that only the forecast VMF1/VMF3 products were utilized for the model evaluation since only these two products could be adapted to real-time GNSS data processing and real-time retrieval of PWV in GNSS meteorology.

**4.1 Using ERA5 hourly reanalysis data as reference**

In this section the ZHD and ZWD obtained from the 3-year (2020–2022) ERA5 hourly reanalysis data at nine pressure levels

at the grid points were used as the reference. The reference ZHD was obtained from the Saastamoinen model, while the ZWD was obtained from using discrete integration, see (Sun et al., 2021a).

Figure 3(a) shows each scheme's mean RMSE of ZHD interpolated from VMF1 and VMF3 at each pressure level in the 3-year period at all global grid points. We can see that Scheme 3 (red) outperformed Schemes 1 and 2 at all levels, and Scheme 2 outperformed Scheme 1 at the three levels of 1000, 900 and 800 hPa. However, the Scheme 2 results were poor at high-

altitude pressure levels, especially in the range above 600 hPa. These results suggest that Scheme 3, which is our new model result, is the best performer for reducing the VMF1/VMF3-based ZHDs.

Figure 3(b) shows the statistical results of the interpolated ZWD values. We can see that Scheme 3 outperformed Scheme 1 and 2 in general, especially in the bottom half range (from 1000 to 600 hPa), where the water vapor content mainly concentrates, and in this range both Schemes 1 and 2 exhibited large differences and Scheme 2 is worse than Scheme 1.

Whilst both had little differences in the upper half range. These results suggest that our new model is the best one for reducing the ZWD obtained from the grid-wise VMF1/VMF3 products, while Scheme 2 may offer an improved accuracy result but only at some pressure levels in comparison with Scheme 1, the officially recommended one.



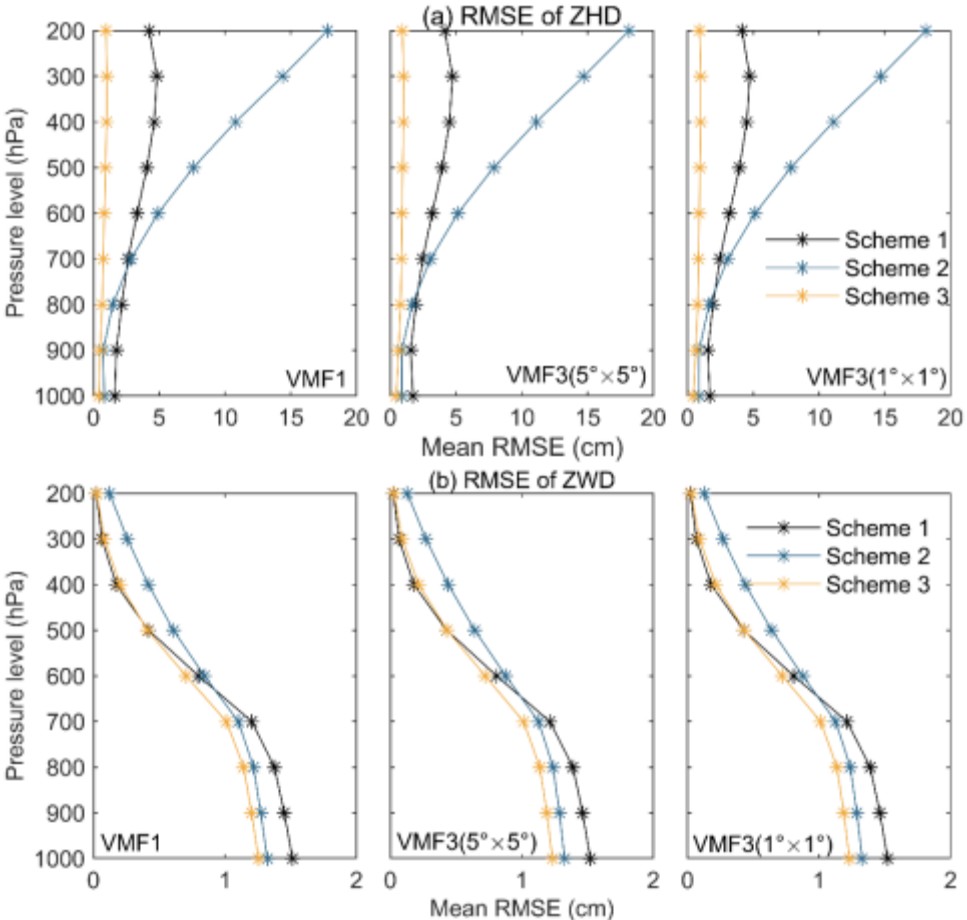

**Figure 3** Mean RMSE of the (a) ZHD and (b) ZWD interpolated from VMF1_FC and VMF3_FC for each scheme at each pressure level in the 3-year (2020 – 2022) period at all global grid points

**4.2 Using IGS ZTD data as reference**

In this section, ZTD values at 394 IGS stations in the 3-year period, same as in the last section, were used as the reference of the ZTD interpolated using the VMF1/VMF3 products for the performance evaluation of the three schemes.

Figure 4 shows the time series of VMF1/VMF3 predicted ZTD errors at IQQE station compared with IGS final ZTD product. As is shown in the figure, the ZTD predicted by Scheme 3 was significantly improved, especially for VMF3(5°×5°). This indicates that the ZTD vertical reduction model has a significant impact on the ZTD interpolation.

Table 2 lists mean, maximum and minimum values of RMSE. As shown in the table, large RMSEs were also found from Schemes 1 and 2. For Scheme 2 (blue), an extreme RMSE value (29.67 cm) occurred at the IQQE station using the VMF3 (5°×5°) product. The maximum RMSE of Scheme 3 was significantly reduced, and the Scheme 3 result suggests the effectiveness of the proposed approach in the improvement of the accuracy of the interpolated ZTD. It should be noted that the IGS ZTD values are ground surface ZTDs, unlike that in Figure 3, the mean RMSE of Scheme 3 had a small

improvement, as most IGS stations are in flat areas. However, for the stations with substantial height disparities from adjacent VMF grid points, the accuracy of the interpolated ZTD was significantly improved. It is expected that, in the ground
surface domain, the new model adds good value to related fields such as GNSS positioning and GNSS meteorology for better performance in regions with large topographic relief.

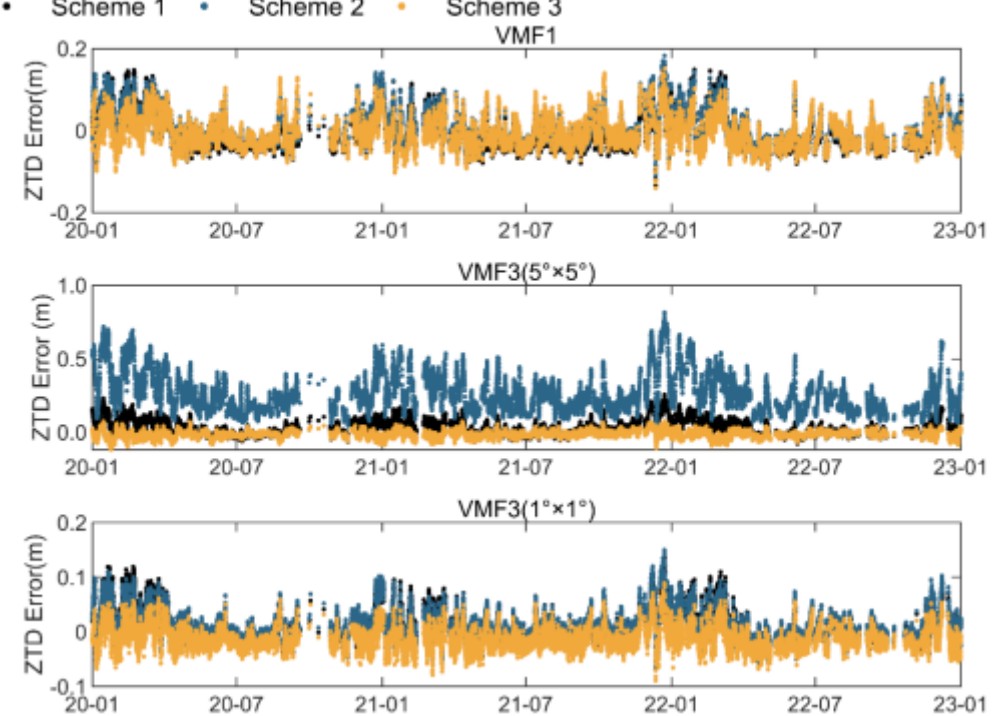

**Figure 4** Time series of VMF1/VMF3 predicted ZTD errors at IQQE station compared with IGS final ZTD product

**Table 1** Mean, minimum and maximum (in cm) of the RMSEs of the ZTD results interpolated for each scheme.

| Schemes | VMF1 | | | VMF3(5°×5°) | | | VMF3(1°×1°) | | |
|---------|------|------|------|------|------|------|------|------|------|
| | min | mean | max | min | mean | max | min | mean | max |
| 1 | 0.51 | 1.90 | **6.79** | 0.90 | 2.20 | **6.05** | 0.80 | 1.60 | **3.16** |
| 2 | 0.69 | 1.82 | **8.25** | 0.52 | 2.21 | **29.67** | 0.52 | 1.55 | **5.45** |
| 3 | 0.76 | 1.78 | **3.89** | 0.61 | 2.05 | **4.45** | 0.62 | 1.51 | **2.84** |

## 5 Conclusions

The ZHD and ZWD provided by grid-based VMF1 and VMF3 tropospheric products are for ground surface values at each of the global grid points, and these products have been widely used for interpolating the *a priori* ZHD and ZWD for GNSS receivers and VLBI stations. In the case that the height of the target GNSS receiver differs largely from its four surrounding
grid points to be used for the interpolation of ZTD, the ZHD and ZWD values at the grid points need to be reduced to the





height of the GNSS station before a horizontal interpolation is performed. Although some simple reduction models are available, their accuracy may not be as good as desired. Thus, in this contribution, new ZHD and ZWD reduction models for each of the four grid points to be used for interpolation were developed to improve the accuracy of interpolated results. The sample data for the modeling were the ZHD and ZWD profiles over the grid points obtained from ERA5 monthly-mean

reanalysis data during the period of 2010–2019. The two sets of reference data used to evaluate the new models were the ZHD and ZWD at nine pressure levels of ERA5 hourly reanalysis data and surface ZTD at 394 globally distributed IGS stations during the 3-year period 2020–2022. Test results showed that the accuracy of the ZHD, ZWD, as well as ZTD interpolated from the VMF1/VMF3 products deduced by the new model was significantly better than the traditional method. The new model is expected to be applied to related fields such as GNSS positioning and GNSS-meteorology for better

performance.

**Acknowledgments**

This work was funded by the National Natural Science Foundation of China (Grant No. 42361134583, 42274021, 42304015), the Jiangsu Provincial Excellent Postdoctoral Program (Grant No. 2023ZB249), the Construction Program of Space-Air

Ground-Well Cooperative Awareness Spatial Information Project (Grant No. B20046), the Independent Innovation Project of "Double-First Class" Construction (Grant No. 2022ZZCX06), the 2022 Jiangsu Provincial Science and Technology Initiative—Special Fund for International Science and Technology Cooperation (Grant No. BZ2022018). The authors would like to thank the ECMWF, TU Wien and IGS for providing ERA5 reanalysis data, grid-based VMF1/VMF3 products and station-wise ZTD products, respectively.


**Competing Interests**

The contact author has declared that none of the authors has any competing interests.

**Data and Code Availability:**

The model developed in this contribution is available at: http://doi.org/10.5281/zenodo.13826894 (Sun, 2024) and GitHub: https://github.com/PengSun1991/VMF_ZTDLpsR, under the MIT License; VMF1 and VMF3 products are available at: http://doi.org/10.17616/R3RD2H(Re3data.Org, 2016); ERA5 monthly mean reanalysis data are available at : https://doi.org/10.24381/cds.6860a573 (Hersbach et al., 2023b); ERA5 hourly reanalysis data on pressure levels are available at: https://doi.org/10.24381/cds.bd0915c6 (Hersbach et al., 2023a); IGS ZTD data are available at NASA Crustal Dynamics

Data Information System (CDDIS) (International GNSS Service, 2000)

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
