# Peer review of "A New Vertical Reduction Model for Enhancing the Interpolation Accuracy of VMF1/VMF3 Tropospheric Delay Products"

_Geoscientific Model Development, 2024_

## Referee Comment (RC1)

**Comments**

The developed a new vertical lapse-rate model to enhance the performance of VMF1/VMF3-based ZHD and ZWD interpolation. Generally speaking, the manuscript is well-written, but some minor revisions may improve the quality of the paper.

[1] Since the tropospheric delays should be mitigated in many space observations, "in GNSS applications" can be removed from the article title, as the model's application scope can be broader.

[2] The authors provided a detailed introduction to the accuracy and applications of VMF1/VMF3 ZTD, but does not seem to address the urgency of improving the accuracy in the introduction part. Thus, the authors are encouraged to revise the introduction part.

[3] Is it necessary to introduce Equation (7) if it is not used in the modeling and evaluation?

[4] "Since the horizontal resolution of the reference data coincides with the VMF1 and VMF3 products, temporal interpolation and horizontal geospatial interpolation were not needed to carry out for the model evaluation." Should be corrected to "Since the temporal and horizontal resolution of the reference data coincides with the VMF1 and VMF3 products, temporal interpolation and horizontal geospatial interpolation were not needed to carry out."

[5] The time resolution, accuracy, data availability (percentage of usable data) and quality control of the GNSS ZTD product should be introduced detailly.

[6] What does $\gamma$ in equation (16) mean, please specify.

---

## Author Response (AR1)

**Response**

Dear editor,

We appreciate the time and effort you and the anonymous reviewers have dedicated to reviewing our work. We have carefully considered each point raised and have made substantial revisions to the manuscript accordingly. A detailed response to the comments is provided below.

**RC#1**

*They developed a new vertical lapse-rate model to enhance the performance of VMF1/VMF3-based ZHD and ZWD interpolation. Generally speaking, the manuscript is well-written, but some minor revisions may improve the quality of the paper.*

*[1] Since the tropospheric delays should be mitigated in many space observations, "in GNSS applications" can be removed from the article title, as the model's application scope can be broader.*

**Response:** We appreciate this insightful comment and agree that the model's application extends beyond GNSS. As tropospheric delay affects various space-based geodetic techniques, including GNSS, DORIS, VLBI, and InSAR, we have revised the title to reflect this broader applicability. The new title is "A New Vertical Reduction Model for Enhancing the Interpolation Accuracy of VMF1/VMF3 Tropospheric Delay Products"

*[2] The authors provided a detailed introduction to the accuracy and applications of VMF1/VMF3 ZTD, but does not seem to address the urgency of improving the accuracy in the introduction part. Thus, the authors are encouraged to revise the introduction part.*

**Response:** This is a good suggestion. We have revised the manuscript, first, we emphasized the importance of ZHD modeling in GNSS data processing: "Given the distinct dynamic characteristics of the ZHD and ZWD, GNSS data processing typically involves correcting for the ZHD while treating the ZWD as an unknown parameter. As hydrostatic and wet mapping functions differ, the error in ZHD cannot be absorbed in the estimated ZWD, which in turn affects the accuracy of ZTD and station height estimations. (Boehm et al., 2006; Tregoning and Herring, 2006; Kouba, 2009). Furthermore, an accurate ZHD is necessary for converting ZTD to precipitable water vapor (PWV) in GNSS meteorology (Bevis et al., 1992; Wang et al., 2017; Zhu et al., 2024), a 1 cm error in ZHD corresponds to a 1.5 mm error in the retrieved PWV."

Second, we presented the urgency of improving the accuracy of VMF1/VMF3 products: "While ZTD derived from VMF1/VMF3 products generally exhibit superior accuracy compared to those derived from empirical tropospheric models, discrepancies in ZHD and ZWD have been documented in certain studies. Specifically, the RMSE of ZHD estimated by grid-wise VMF1/VMF3 using the recommended interpolation method can reach 5 cm when compared with reference ZHD values obtained from radiosonde measurements in some regions. Similarly, the

RMSE of ZWD can also attain substantial magnitudes. These findings underscore the potential limitations of currently widespread methods under specific geographical or atmospheric conditions (Sun et al., 2021b, a)."

*[3] Is it necessary to introduce Equation (7) if it is not used in the modeling and evaluation?*

**Response:** We re-visited the manuscript, and this equation has been deleted in the new manuscript.

*[4] "Since the horizontal resolution of the reference data coincides with the VMF1 and VMF3 products, temporal interpolation and horizontal geospatial interpolation were not needed to carry out for the model evaluation." Should be corrected to "Since the temporal and horizontal resolution of the reference data coincides with the VMF1 and VMF3 products, temporal interpolation and horizontal geospatial interpolation were not needed to carry out."*

**Response:** Amended.

*[5] The time resolution, accuracy, data availability (percentage of usable data) and quality control of the GNSS ZTD product should be introduced detailly.*

**Response:** This is a good suggestion. The manuscript has been revised: "A rigorous quality control procedure was implemented to ensure the quality of the reference ZTDs. To mitigate the impact of known midnight discontinuities present in the IGS ZTD time series, only odd-numbered UTC epochs (i.e., 1, 3, ..., 23) were retained, thus avoiding potential offsets in the reference data. Initially, the IGS ZTD time series, originally at 5-minute intervals, was resampled to a 2-hour interval. Subsequently, epochs with a standard deviation exceeding 4 mm, as indicated within the IGS ZTD products, were excluded. Finally, following these two filtering stages, stations with fewer than 5000 remaining ZTD epochs were removed from consideration to capture the seasonal variation of the tropospheric delay."

[6] What does $\gamma$ in equation (16) mean, please specify.

**Response:** Amended. $\gamma$ is the ZWD decay parameter defined by Dousa. See:

DOUSA J, ELIAS M. An improved model for calculating tropospheric wet delay[J]. Geophysical Research Letters, 2014, 41(12): 4389-4397.

**RC#2**

*The paper addresses a critical aspect of GNSS applications. The proposed new model for ZTD was significantly better than the ones deduced by traditional methods using ERA5 and IGS-ZTD as reference. However, I'm curious if the new model could hold significant promise for enhancing GNSS positioning accuracy. With additional details on methodology, expanded validation, the work could set a strong foundation for practical implementation. I recommend the paper for publication with minor revisions.*

*Detail Comments:*

*1. Due to this study focuses on GNSS applications, I suggest to add the experiment for the application of the new model in the GNSS navigation. The new model could be also assessed more comprehensively, which could further highlight the significance of the new model enhancing the GNSS navigation.*

**Response:** This is a very good suggestion. We acknowledge the reviewer's valuable suggestion regarding the application of the proposed model within GNSS navigation. We concur that such an evaluation is crucial for demonstrating the model's practical utility.

Tropospheric delay constitutes a significant error source in GNSS code and phase observations, directly impacting positioning accuracy. As described by Equation (5) in the manuscript, the hydrostatic and wet components of the tropospheric delay are modeled using distinct mapping functions. At low elevation angles, the divergence between these mapping functions becomes pronounced, leading to hydrostatic/wet mapping separation errors that affect GNSS-derived station heights and ZTD estimations.

For precise GNSS positioning, ZWD is usually estimated as a time-varying unknown parameter, while ZHD is corrected directly. Studies have demonstrated that, with a 5-degree elevation cutoff, hydrostatic/wet mapping separation errors can induce height errors approximately one-tenth the magnitude of the ZHD error. Consequently, achieving 1 mm height accuracy necessitates ZHD accuracy on the order of 10 mm. In standard Single Point Positioning (SPP), where ZWD is often corrected directly, inaccuracies in ZWD also propagate into the estimated coordinates.

Furthermore, accurate ZHD modeling is essential for GNSS meteorology applications. As previously discussed, ZHD influences the accuracy of ZTD estimation. Moreover, ZHD directly affects the retrieval of Precipitable Water Vapor (PWV) from ZWD. Assuming accurate ZTD estimation, a 1 cm error in ZHD translates to a 1.5 mm error in the retrieved PWV.

Based on the above discussion, we conclude that existing research has sufficiently demonstrated the impact of improved tropospheric modeling accuracy on GNSS applications. This research has validated the improvement achieved by the proposed model in tropospheric delay prediction, with a magnitude significant enough to exert a substantial influence on GNSS positioning and GNSS meteorology. The new model is very easy to implement, however, a sophisticated precise point positioning (PPP) software package, developed in C++ and incorporating the proposed model, is currently under development. Potential issues in other modules may temporarily hinder the comprehensive evaluation of the model's performance. We intend to thoroughly investigate the reviewer's suggestion following the release of a stable software version. In the interim, a MATLAB implementation of the newly developed model has been made publicly accessible for testing and evaluation purposes.

*2. The Introduction has reviewed the detailed development of "VMF1/VMF3 ZTD". However, it has no information about why we need "A New Reduction Model for Enhancing the Interpolation Accuracy of VMF1/VMF3 Tropospheric Products in GNSS applications". If the officially recommended ZTD interpolation method is enough accurate, a new vertical reduction model may be not making much sense.*

**Response:** This is a very good suggestion. We have revised the manuscript to emphasized the importance of developing a new lapse rate model for the grid-wise VNMF1/VMF3 model. "While ZTD derived from VMF1/VMF3 products generally exhibit superior accuracy compared to those derived from empirical tropospheric models, discrepancies in ZHD and ZWD have been documented in certain studies. Specifically, the RMSE of ZHD estimated by grid-wise VMF1/VMF3 using the recommended interpolation method can reach 5 cm when compared with reference ZHD values obtained from radiosonde measurements in some regions. Similarly, the RMSE of ZWD can also attain substantial magnitudes. These findings underscore the potential limitations of currently widespread methods under specific geographical or atmospheric conditions (Sun et al., 2021b, a).".

*3. There are many reanalysis data. Why do you choose ERA5 for the development of the new ZTD vertical reduction model?*

**Response:** This is a very good suggestion. Many countries and organizations are dedicated to developing high-quality reanalysis datasets. Examples include NCEP/NCAR and MERRA-2 from the United States, ERA5 from Europe, JRA-55 from Japan, and CRA40 from China, etc. Each of these datasets has its own strengths and characteristics. We chose to use ERA5 data in our research. While ERA5 is widely recognized for its excellent data quality, our primary reason for this choice is its consistency with our research subject: the VMF1/VMF3 grid products released by TU Wien. These products are also based on data from the European Centre for Medium-Range Weather Forecasts (ECMWF), the same source as ERA5. Using ERA5 ensures better consistency between our newly developed model and the VMF1/VMF3 grid products. Although we haven't yet compared the modeling results using other datasets, we plan to explore these options in future research."

*4. For the GNSS ZTD data do you have done quality control? There may data gaps or jumps in the data, which strategy do you use for them?*

**Response:** This is a very good question. IGS ZTD data have some gaps/jumps, which may affect the evaluation results. We apologize for missing the quality control information in the original manuscript, and here is a revised one: "A rigorous quality control procedure was implemented to ensure the quality of the reference ZTDs. To mitigate the impact of known midnight discontinuities present in the IGS ZTD time series, only odd-numbered UTC epochs (i.e., 1, 3, ..., 23) were retained, thus avoiding potential offsets in the reference data. Initially, the IGS ZTD time series, originally at

5-minute intervals, was resampled to a 2-hour interval. Subsequently, epochs with a standard deviation exceeding 4 mm, as indicated within the IGS ZTD products, were excluded. Finally, following these two filtering stages, stations with fewer than 5000 remaining ZTD epochs were removed from consideration to capture the seasonal variation of the tropospheric delay.".

---

## Author Response (AR2)

Dear editor,

We sincerely appreciate your valuable comments and constructive suggestions. We have carefully addressed all the points raised and have revised the manuscript accordingly.

• *Specify the units for all variables in the manuscript (e.g., TD, N, s, L, …).*
**Response**: Added.

• *L60: "As hydrostatic and wet mapping functions differ": What do you mean? Please reword.*
**Response**: The slant tropospheric delay, as is shown in equation (5), can be expressed as:

$$TD = ZHD \cdot mf_h + ZWD \cdot mf_w + \Delta T_{grad} \tag{5}$$

where $mf_h$ and $mf_w$ are the mapping functions for the hydrostatic (ZHD) and non-hydrostatic (ZWD) part of the tropospheric delay. "As hydrostatic and wet mapping functions differ" means that $mf_h$ and $mf_w$ have different values. When fixing ZHD to estimate ZWD, since there is a difference between $mf_h$ and $mf_w$, the ZHD error will not be completely absorbed by the estimated ZWD, which in turn affects the accuracy of the ZTD estimation (the sum of ZHD and ZWD).

• *L73: "in-site meteorological … water vapor": This sentence seems contradictory. Why do in-situ meteorological measurements fail to measure dynamic nature of water vapor?*

**Response:**

Sorry for using such a seemingly contradictory expression.

The original draft of this sentence is "Although the ZWD can also be calculated in a similar way, i.e., using an empirical model, e.g., the Askne-Nordius model (Askne and Nordius, 1987), together with in-situ meteorological measurements, its accuracy is not as good as that of the Saastamoinen model due to the dynamic nature of water vapor (Chen and Liu, 2016)."

Explanation: Askne-Nordius model is an empirical model for calculating ZWD, which needs in-situ water vapor pressure as input parameter. The water vapor pressure can be measured by in-situ meteorological sensors, however, the Askne-Nordius model itself is not a very accurate model because the water vapor changes rapidly in both spatial and temporal domain.

The revised version is "Although the ZWD can also be calculated using empirical models such as the Askne-Nordius model (Askne and Nordius, 1987), which relies on in-situ meteorological measurements (e.g., water vapor pressure), its accuracy is generally lower than that of the Saastamoinen model. This is because water vapor exhibits high spatiotemporal variability, and the Askne-Nordius model's empirical formulation cannot fully capture these rapid fluctuations, even when precise in-situ measurements are available (Chen and Liu, 2016)."

• **L76-77: Provide the official names of NWM, UNB3m, and GPT.**

Response:

NWM: numerical weather model;

UNB3m is one version of UNB Neutral Atmosphere Delay Model, where UNB denotes University of New Brunswick, "3m" is the number of the model version.

GPT: Global Pressure and Temperature model.

The revised text is: "*Since most GNSS stations are not mounted with meteorological sensors and it is complex for real-time GNSS users to process forecasted NWM (numerical weather model) data to obtain the atmospheric parameters, empirical tropospheric delay models like UNB3m (Leandro et al., 2006) developed by University of New Brunswick (UNB) and GPT (Global Pressure and Temperature) models developed by Vienna University of Technology (TU Wien)*"

**• L78-81: I am unsure about the references to empirical models here, maybe short review for the studies might be necessary for explaining limitations of previous studies.**

**Response:** The manuscript has been revised:

"In such empirical models, spatiotemporal variations of the atmospheric parameters are modelled, and then the atmospheric parameters can be predicted directly. Incorporating advanced height correction model is an effective method to improve the modeling accuracy of atmospheric parameters or the tropospheric delays (Huang et al., 2023; Jiang et al., 2024; Li et al., 2018; Sun et al., 2023). However, while easy to use, these models have limited accuracy due to rapid variation of the troposphere (Wang et al., 2017; Xia et al., 2023), as these models can only capture the mean status of the annual, semi-annual and diurnal variations of the parameters."

**• L90: Provide the official names of GFZ-VMF1 and UNB-VMF1.**

Response: Both GFZ and UNB are institution names. We have revised the manuscript: "In addition, some other VMF1-like products are also publicly available for users (Santos, 2011; Zus et al., 2015)."

**• L95: "a difference of 1.83 cm mean root-mean-square"?**

**Response:** corrected.

**• L108-113: Provide the references for each sentence.**

**Response:** While ZTD derived from VMF1/VMF3 products generally exhibit superior accuracy compared to those derived from empirical tropospheric models, discrepancies in ZHD and ZWD have been documented in certain studies (Sun et al., 2021b; Yang et al., 2021; Yao et al., 2018b). Specifically, the RMSE of ZHD estimated by grid-wise VMF1/VMF3 using the

recommended interpolation method can reach 5 cm when compared with reference ZHD values obtained from radiosonde measurements in some regions (Sun et al., 2021b). Similarly, the RMSE of ZWD can also attain substantial magnitudes (Sun et al., 2021a; Yang et al., 2021).

**• L111: "Similarly, … magnitudes" Provide the reason for enlargement of "the RMSE of ZWD".**

**Response:** This is a good question.

Firstly, the accuracy of the source data for the VMF1/VMF3 grid products—namely, the atmospheric profile outputs from ECMWF's numerical weather models—is not uniformly distributed globally. Secondly, since the VMF1/VMF3 grid products only provide surface values at grid points, when there is a significant elevation difference between the target location and the surrounding grid points, an imprecise ZWD vertical correction model can lead to substantial interpolation errors in ZWD.

**• L113: Specify the geographical and atmospheric conditions that discrepancies in ZHD and ZWD occur.**

**Response:** This is a very good question too. As is mentioned above, since the VMF1/VMF3 grid products only provide surface values at grid points, when there is a significant elevation difference between the target location and the surrounding grid points, an imprecise ZHD/ZWD vertical correction model can lead to substantial interpolation errors in ZHD/ZWD.

**• L133: "394 IGS stations were selected" -> out of how many sites?**

**Response:** The IGS routinely produce ZTD products of the IGS stations. But the number of the stations provided is not fixed. For example, the number for 31/12/2022 is 418, for 01/01/2020 is 382. ZTD products in the 3-year period of 2020–2022 at 394 IGS stations were selected using the criteria mentioned in the manuscript.

**• L137: Provide the method how 5-min time series were resampled to 2-hr ones.**

**Response:** The IGS ZTD product provide ZTD values at fixed epochs with 5 min interval: 00:00, 00:05, 00:10…01:00…, thus we can resample the data directly by picking up specific epochs.

**• L156: ZHD0 in (8) and ZWD0 in (10) -> ZHD0 in (7) and ZWD0 in (9)**

**Response:** Corrected.

**• L158, 165, 192, 228: Saastamoinen model -> Eq. (6)**

**Response:** Corrected.

**• L163: What do you intend by "S (A'~D')"? Consider rewording this sentence.**

Response: As is shown in Figure 2, Points A, B, C, and D are the surface grid points of VMF1/VMF3, and S is the target location. The horizontal plane passing through S intersects the vertical lines of A, B, C, and D at points A'–D', meaning that the elevation of S is the same as that of A'–D'.   First, we need to correct the tropospheric delay values at A, B, C, and D along the elevation direction to obtain the tropospheric delay values at A'–D'. Then, we perform horizontal interpolation to derive the tropospheric delay parameters at point S.

• **L164: What are "target" and "reference"? Add detailed information.**

**Response:** The manuscript has been revised: $\Delta h$ is the difference between the target and reference heights, i.e. $\Delta h_{AA'}$ , $\Delta h_{BB'}$, $\Delta h_{CC'}$ , $\Delta h_{DD'}$

• **L166: What is "product-provided ZWD"? Add explanation.**

**Response:** Product-provided ZWD means the grid-wise ground surface ZWD values provided by VMF1/VMF3 product. The manuscript has been revised:"[6]. Using ZWD0 (i.e., ZWDs at A to D) and the following model to obtain ZWDs for A'~D':"

• **L166: Is "reduction model" correct phrase in English?**

**Response:** The manuscript has been revised:"[6]. Using $ZWD_0$ (i.e., ZWDs at A to D) and the following model to obtain ZWDs for A'~D':"

• **L238: "These results…" I suspect this results as self-evident because both the reference and Scheme 3 (т and г) are estimated using ERA5 data. I recommend applying different reanalysis data for estimates of the reference and Scheme 3.**

**Response:**

This is a very good suggestion. Both the reference and Scheme 3 are based on ERA5 data, however, the reference data are ERA5 hourly data from 2020 to 2022, while the Scheme 3 (an empirical model for lifting surface ZTD to the target height) was developed using ERA5 monthly mean data from 2010 to 2019. The VMF1/VMF3 surface ZTD product are developed using ECMWF OPERATIONAL NWM data and FORECAST NWM data. The most important reason for using ERA 5 in the previous manuscript is that the error embedded in VMF1/VMF3 surface ZTD product is also a major error source when interpolating ZTD at the target position, using reanalysis data provided by ECMWF may eliminate the discrepancy resulting from other reanalysis data, such as MERRA-2 and JRA55.

To further evaluate the performance of these three methods, radiosonde data from 2020 to 2022 at 608 stations are tested and the results demonstrates similar conclusions, i.e., the new model developed in this research is a good method for improving the ZTD interpolation accuracy of VMF1/VMF3 surface ZTD product. The manuscript has been revised.

• **L 248: What is "IQQE station"? Why is this station selected for the analysis? Does this**

**station guarantee representativeness of 394 IGS station? Provide detailed explanation.**

**Response:**

This is a good question.

*What is "IQQE station"?* IQQE is the name of an IGS station.

*Why is this station selected for the analysis?* The ZTD time series of this station was selected to prove that for the stations with substantial height disparities from adjacent VMF grid points, the accuracy of the interpolated ZTD can be significantly improved by utilizing the new model proposed in this paper.

*Does this station guarantee representativeness of 394 IGS station?* When interpolating *a priori* ZTD at the GNSS station using VMF1/VMF3 product, the height differences between the station and the adjacent VMF grid points should be considered. Such height differences are small for part of these stations, which means that the scheme 1 (official one) may not leads to large ZTD prediction errors. However, for the stations with substantial height disparities from adjacent VMF grid points, the accuracy of the interpolated ZTD can be significantly improved by the new method, such as IQQE station presented in the manuscript. In fact, there are thousands of other geodetic GNSS stations running on the earth surface, continuously observing signals broadcast by GNSS satellites (GPS, Galileo, QZSS, etc.), and large amount of civilian GNSS receivers are mounted on rover objects like Unmanned Aerial Vehicle (UAV), cars, and mobile phones, this implies that large elevation differences between the GNSS receiver and the VMF1/VMF3 grid points are highly likely.

---

## Author Response (AR3)

Dear editor,

We sincerely appreciate your time and efforts in handling our manuscript. We have carefully considered all the comments and revised the manuscript accordingly. Our point-by-point responses are provided below:

Response:

**1) First, I recommend the authors to edit the manuscript by using professional proofreading service before resubmitting the revision.**

**Response:** Thank you for your constructive suggestion. We have thoroughly revised the manuscript with the assistance of a native English speaker (from Australia) with experience in academic writing. We kindly ask you to assess whether the current language quality meets the standards for publication, and we are happy to make further improvements if necessary.

**2) L76: Reword to "numerical weather model (NWM)".**

**Response:** Revised as suggested.

**3) L77: Reword to "Global Pressure and Temperature (GPT)".**

**Response:** Revised as suggested.

**4) L79: Following sentences are too vague. "In such empirical models, spatiotemporal variations of the atmospheric parameters are modelled, and then the atmospheric parameters can be predicted directly. Incorporating advanced height correction model is an effective method to improve the modeling accuracy of atmospheric parameters or the tropospheric delays (Huang et al., 2023; Jiang et al., 2024; Li et al., 2018; Sun et al., 2023). However, while easy to use, these models have limited accuracy due to rapid variation of the troposphere (Wang et al., 2017; Xia et al., 2023), as these models can only capture the mean status of the annual, semi-annual and diurnal variations of the parameters."**

**Here do you intend the following contents?**

**"For instance, some studies improved the prediction of the atmospheric parameters by incorporating an advanced height correction in these empirical models (e.g., Li et al., 2018; Huang et al., 2023; Sun et al., 2023; and Jiang et al., 2024). Nevertheless, their results are still insufficient in accuracy of WHAT (Temperature? Pressure? Moisture? Boundary-layer height? Or other variables?) due to coarse spatiotemporal resolution for modelling those rapid variations in the troposphere (Wang et al., 2018; Xia et al., 2023)."**

**Authors should revise the sentences with brief and precise description.**

**Response:**

We appreciate for this constructive comment. This paragraph has been revised to enhance the readability, which now reads:

These models operate independently of external meteorological inputs and empirically estimate atmospheric parameters (such as atmospheric pressure, water vaper pressure, temperature, etc.) based solely on a given location and time. The UNB3m model uses lookup tables that provide the mean and annual amplitude of meteorological variables at mean sea level, facilitating tropospheric delay computation through standardized vertical reduction models. Boehm et al. (2007) introduced the first version of the GPT model, which represents global atmospheric pressure and temperature using spherical harmonics. Its successor, GPT2 (Lagler et al., 2013) advanced the GPT series by implementing a global 5°×5° grid and characterizing atmospheric pressure, temperature, temperature lapse rate, and water vapor pressure by accounting for their mean, annual, and semi-annual harmonics. The GPT2w model (Böhm et al., 2015) refined this framework by incorporating additional parameters and increasing the resolution to 1°×1°. The GPT3 (Landskron and Böhm, 2018) integrated an empirical gradient model while maintaining the other meteorological parameters consistent with GPT2w. Both the UNB3m and GPT model series furnish meteorological parameters at a single reference level (either mean sea level or Earth's surface), necessitating their vertical propagation to the desired elevation. To enhance the accuracy of tropospheric delay modelling, recent studies have introduced more advanced modelling techniques that better describe the height-dependent variability of atmospheric parameters (Huang et al., 2023; Jiang et al., 2024; Li et al., 2018; Sun et al., 2023). Nevertheless, while these empirical models can predict atmospheric parameters with reasonable accuracy, they are fundamentally limited to capturing long-term average variations, primarily annual and semi-annual cycles. As a result, their predictive accuracy is inherently constrained by the atmosphere's continuous and often abrupt variability, particularly for rapidly fluctuating parameters such as air temperature and water vapor pressure (Wang et al., 2017; Xia et al., 2023)

Please let us know if further revision or clarification is needed.

**5) L129-133: Authors should provide key references.**

**Response:** Thank you for the suggestion. We have added two relevant references to support Equation (4). Regarding the criterion that each station must have at least 500 profiles during the experimental period, this was defined to ensure data reliability and consistency. Including stations with too few profiles could introduce biases and reduce the robustness of the statistical evaluation.

**6) Table 1: Authors wrote the "Table 1 lists … found from Schemes 1 and 2" in L251, but**

**Table 1 provides information on only difference among VMFs and no information on schemes. Authors should provide all necessary information in Table 1 and describe the results in the manuscript.**

**Response:** We appreciate the reviewer's careful reading and acknowledge the confusion caused by our previous wording. The description of "Schemes 1, 2, 3" in the manuscript refers to the experimental setups introduced at the beginning of Section 4:

*4 Results and discussion*

*To compare the accuracies of the standard and alternative vertical reduction models for reducing the ground surface ZHD and ZWD from the grid reference height to GNSS station heights, the following three schemes were tested using ZHD and ZWD data obtained from grid-wise VMF1/VMF3 products.*

*1) Scheme 1: for the officially recommended reduction methods, which utilizes fixed empirical decay parameters, corresponding to Eq. (9) for ZHD and Eq. (10) for ZWD.*

*2) Scheme 2: for the temperature-dependent pressure decay model (Eq. (11)) and the exponential ZWD decay model (Eq. (12)). The required atmospheric variables, including temperature ($T_0$), temperature lapse rate (β), and water vapor decay coefficient (λ), are predicted by the GPT2w model.*

*3) Scheme 3:   for the new vertical reduction model developed in this research, i.e., Eq. (15) for ZHD and Eq. (18) for ZWD.*

**7) L252: Do not describe explanation of Fig. 4 without specifying the figure.**

**Response:** Thank you for pointing this out. We have revised the paragraph to explicitly refer to Figure 4 at the beginning of the explanation, thereby improving the clarity and structure of the discussion.

**8) L252: IQQE IGS station is first appearance here. Authors should provide basic information on this station and reason why this station was selected for reference data.**

**Response:**

Thank you for this constructive suggestion. We have revised the manuscript to introduce the IQQE station with appropriate context.

We selected the IQQE IGS station as a key example to demonstrate the significant impact of height differences between a GNSS station and its neighboring VMF (Vienna Mapping Function) grid points on the interpolation accuracy of tropospheric delay. IQQE (IGS) station is located in Iquique, Chile, lies to the west of the Andes Mountains, thus this station exhibits substantial differences in height compared to its surrounding closest VMF1/VMF3 grid points: the maximum height differences reach 1562 m for VMF1, 4632 m for VMF3 (5°×5°), and 2750 m for VMF3 (1°×1°). These significant height disparities create challenging conditions where

the traditional method (referred to as Scheme 1 in our study) tend to produce large biases. By showcasing IQQE, we effectively illustrate how our novel method maintains robust performance even in such complex topographical environments. This highlights the superior accuracy of our approach in mitigating errors caused by height variations.

---

## Author Response (AR4)

**Response**

Dear editor,

We would like to express our sincere gratitude for the time and effort you have dedicated to reviewing our manuscript and providing these constructive comments. We have carefully considered each suggestion and have prepared the following point-by-point responses:

1. *L66: "PWV" term is first appearance. Reword to "OFFICIAL NAME (PWV)".*

Response: Revised. "precipitable water vapor (PWV)"

2. *L70: Reword "The zenith hydrostatic delay (ZHD)" to "The ZHD".*

Response: Revised

3. *L156: "were removes.," -> "were removed,"*

Response: Revised

4. *L159: "(b) 394 GNSS stations": Do you intend to describe "394 IDS stations"?*

Response: Revised. "394 IGS stations"

5. *L269: "the time series of errors in VMF1/VMF3": Define the word "error".*

Response: This paragraph has been revised, and the sentence "the time series of errors in VMF1/VMF3" is deleted now.

6. *L269: Provide information on IQQE station: official name, latlon, and altitude.*

Response: Added. "The IQQE (IGS network) station is located in Iquique, Chile (latitude: -20.273542°, longitude: -70.131717°, height:38.9 m),…"

7. *L247; 264; 265; 267; 269; 275: In general, "significantly" term is used to refer to "statistical significance" that is determined by statistical test p-value. Provide the p-value for each sentence to ensure the scheme 3 superiority.*

Response: This is a very good suggestion. We completely agree that p-value is important to prove the word "significantly". To evaluate the performance of the new model, nearly all available radiosonde stations and IGS stations are utilized, thus we believe that the RMSEs provided in the manuscript has demonstrated the performance of the proposed new model. The word "significantly" used in the previous manuscript is quite confusing, thus we revised all the words "significantly", "significant" to "considerably", "considerable" to improve the readability.

8. *L270: "The figure indicates a significant effect of …": It was unable to identify "significant difference" among three schemes from given Figure 4 showing overlapped dot plots. Consider plotting with different types of graphs, such as scatter graph, and/or be sure to provide statistical information to ensure your description.*

Response: Thanks for this constructive suggestion. We have provided a new figure with statistical information to improve the quality of the manuscript. The new sentences and figure now read:

As an example, Figure 4 shows the correlation and accuracy analysis of ZTD interpolated from VMF1/VMF3 products using three schemes at IQQE station. The IQQE (IGS network) station is located in Iquique, Chile (latitude: -20.273542°, longitude: -70.131717°, height:38.9 m), which lies to the west of the Andes Mountains, thus this station exhibits substantial differences in height compared to its surrounding closest VMF1/VMF3 grid points: the maximum height differences reach 1562 m for VMF1, 4632 m for VMF3 (5°×5°), and 2750 m for VMF3 (1°×1°). As is shown in the figure, S Scheme 3 consistently demonstrated the lowest ZTD RMSE among all three schemes and products. Specifically, for the VMF3 (5°×5°) grid, the ZTD RMSE from Scheme 1 was 6.05 cm, while Scheme 2 resulted in a notably high RMSE of 29.67 cm, and Scheme 3's RMSE was only 2.87 cm. The figure indicates a substantial influence of height differences between a GNSS station and its neighboring VMF1/VMF3 grid points on the interpolated ZTD (ZHD+ZWD), and the new model proposed in this research is strongly recommended in such cases.

[Figure]

**Figure 4** Correlation and accuracy analysis of ZTD interpolated from VMF1/VMF3 products and three schemes at IQQE station

---

## Author Response (AR5)

**Response**

Dear editor,

We appreciate the time and effort you have dedicated to reviewing our manuscript. We have carefully checked all the comments and have revised the manuscript accordingly, see the point-to-point responses followed:

1. *L273: "As is shown in the figure, S Scheme3" -> "As is shown in the figure, Scheme3"*

Response: corrected.

2. *L276: "Scheme 3's RMSE" -> "RMSE value of the Scheme 3"*

Response: revised.